# The Effect of Aging on the ERP Correlates of Feedback Processing in the Probabilistic Selection Task

**DOI:** 10.3390/brainsci10010040

**Published:** 2020-01-09

**Authors:** Robert West, AnnMarie Huet

**Affiliations:** 1Department of Psychology and Neuroscience, DePauw University, Greencastle, IN 46135, USA; 2Department of Psychology, Ohio University, Athens, GA 45701, USA; ah686116@ohio.edu

**Keywords:** aging, feedback processing, feedback negativity, P3, FN

## Abstract

Feedback processing contributes to efficient learning, decision making, and social interaction. Studies using event-related brain potentials (ERPs) reveal that feedback processing is associated with transient ERP components over the medial frontal and posterior regions of the scalp that distinguish between positive and negative feedback. There is some evidence indicating that aging has differential effects on the ERP correlates of feedback processing in a gambling task, and the current study was designed to extend these findings to a reinforcement learning paradigm. Younger and older adults performed the probabilistic selection task while ERPs elicited by feedback cues indicating a correct or incorrect choice were recorded during the learning phase. The ERPs revealed that the amplitude of the feedback negativity and frontal P3 were attenuated in older adults relative to younger adults. The amplitude of a temporal positivity was also attenuated in older adults; in contrast, the amplitude of an occipital negativity was insensitive to the effects of aging. These findings indicate that aging may be associated with the disruption of both local activity and long-range connectivity between neural structures related to feedback processing.

## 1. Introduction

The ability to efficiently process feedback related to the outcome of one’s decisions represents a fundamental aspect of information processing that may be related to reinforcement learning, financial decision making, and social interaction [1,2,3]. Studies using event-related brain potentials (ERPs) consistently reveal two components (i.e., feedback negativity (FN) and frontal P3) over the medial frontal region of the scalp that distinguish positive and negative feedback between 200 and 500 ms after the onset of feedback [4,5]. The FN represents greater negativity for negative feedback than for positive feedback; the frontal P3 follows the FN and represents greater positivity for negative feedback than for positive feedback. The current study sought to examine the generalizability of previous research from our laboratory that revealed a paradoxical effect of aging on the ERP correlates of feedback processing between 300 and 500 ms over the medial frontal region [6]. This effect reflected an age-related reduction in the effect of feedback on the FN, but not the frontal P3, that were both localized to the anterior cingulate cortex (ACC) [6]. These findings led to the suggestion that the effects of aging on the ERP correlates of feedback processing may result from the disruption of inputs to the ACC, rather than from the primary disruption of function within the ACC.

ERPs have been used in a number of studies to examine the neural correlates of feedback processing in simple gambling and reinforcement learning tasks [5]. This literature reveals three components that distinguish between positive and negative feedback. Two over the medial frontal region of the scalp (i.e., FN and frontal P3) and one over the central-parietal region (i.e., parietal P3.). The FN represents greater negativity for negative outcomes than for positive outcomes between 200 and 300 ms after stimulus onset and appears to be more strongly related to the valence than the magnitude of an outcome [4,7]. The frontal P3 represents greater positivity for negative outcomes than for positive outcomes following the FN and is sensitive to both the valence and magnitude of outcomes [8,9,10,11]. While there are likely differences in the psychological variables that contribute to the FN and frontal P3, these two components of the ERPs share a common topography over the medial frontal region of the scalp and have been localized to neural generators in the ACC or medial frontal cortex [12,13]. In contrast to the FN and frontal P3, the parietal P3 is often greater in amplitude for correct feedback than for incorrect feedback [14] or for gains than losses [15].

Two recent studies reveal that the FN and frontal P3 are associated with transient ERPs over the temporal, lateral frontal, and occipital regions of the scalp [13,14]. These modulations of the ERPs were localized to the superior temporal cortex or inferior frontal cortex, inferior temporal gyrus, and posterior cingulate/occipital lobe. This ERP evidence is consistent with work using MEG to examine oscillatory activity related to feedback processing that reveals correlations between frontal and occipital, parietal, and temporal regions in beta-band power [16]. Also, converging with this electophysiological evidence, studies using functional magnetic resonance imaging (fMRI) reveal that feedback processing is related to a network that includes the ACC, posterior cingulate, striatum, orbital frontal cortex, and cerebellum [17,18].

The effects of aging on the FN and frontal P3 related to feedback processing have been examined in studies using both reinforcement learning paradigms [19,20,21] and gambling tasks [6]. The effect of aging on the FN is fairly uniform across studies, reflecting a decrease in the difference in amplitude between the ERPs for negative and positive feedback in older adults relative to younger adults [20,21,22]. In some studies, the effect of aging reflects an attenuation of the FN in older adults [22], while in other studies the amplitude of the FN is similar for negative and positive feedback in older adults [19]. These findings lead to the suggestion that aging is associated with a decline in the functional integrity of the ACC/medial frontal cortex, resulting in the disruption of processing negative feedback.

The effect of aging on the frontal P3 related to feedback processing has been examined in fewer studies than is the case for the FN. West et al. [6] reported that the amplitude of the frontal P3 was similar in younger and older adults in a gambling task, and a similar outcome may exist for reinforcement learning. For instance, in a spatial learning task the age by outcome interaction was not significant for the P3, while the FN was essentially absent in older adults [22]. Similarly, in Eppinger et al. [19] the amplitude of the frontal P3 following the FN appears to be greater for negative feedback than for positive feedback; however, the frontal P3 was not examined in the data analyses, so it is not possible to know whether there was a significant difference between younger and older adults. In contrast to the effect of aging in the FN, the absence of an effect of aging on the frontal P3 leads to the suggestion that the functional integrity of the ACC/medial frontal cortex may be intact in later adulthood. Together, differences in effects of aging on the FN and frontal P3 lead to the hypothesis that age-related differences in feedback processing may result from the disruption of input to ACC/medial frontal cortex from structures within the lateral frontal, temporal, and occipital cortices, rather than primary impairment within the ACC.

Given the contrasting effects of aging on the FN and frontal P3 within the context of feedback processing, the current study had two goals. First, we sought to provide a conceptual replication and extension of the findings of West et al. [6] using a reinforcement learning task to establish the generalizability of the paradoxical effects of aging on the FN (decline) and frontal P3 (stability) observed in a gambling task. Second, we sought to examine the effects of aging on transient ERP components located outside of the medial frontal cortex (i.e., over the temporal and occipital regions of the scalp) that covary in time with the FN and frontal P3 [13,14]. If the effect of aging on the FN results from a decline in input from lateral cortical structures [6], we predicted that the amplitude of temporal positivity would be reduced in older adults relative to younger adults. In contrast, we predicted that the occipital negativity would not reveal an effect of age, if input from outside the medial frontal region contributing to the frontal P3 is preserved in later adulthood. In anticipation of the results, the hypotheses regarding the FN and temporal positivity were supported; and the amplitude of the frontal P3, but not the occipital negativity, was attenuated, providing partial support for the hypotheses related to these two ERP components.

## 2. Materials and Methods

### 2.1. Participants

Younger adults were recruited from a participant pool maintained by the Department of Psychology at Iowa State University, and community-dwelling older adults were recruited from the membership of a lifelong learning program affiliated with the university. Thirty-seven younger adults (ages 18–24; females = 14) and 36 older adults (ages 63–85; females = 23) provided usable behavioral and electroencephalogram (EEG) data for the study. A chi-square test including age and gender revealed that there were significantly more females in the older adult group, X^2^(1) = 4.50, *p* = 0.034. The data for eight younger adults were not included due to equipment failure during data collection or poor quality of the EEG, and the data for one older adult were not included as they reported closing their eyes during the transfer phase of the probabilistic selection task (PST). The demographic and psychometric data for the two groups are presented in Table 1. The younger adults had fewer years of education than the older adults, *t*(68) = −6.85, *p* < 0.001, d = −1.64, and the older adults scored higher on the vocabulary test, *t*(71) = −8.56, *p* < 0.001, d = −2.01. The older adults completed fewer items on the digit symbol test than the younger adults, *t*(71) = 7.05, *p* < 0.001, d = 1.65.

### 2.2. Materials

Probabilistic selection task. In the learning phase of the PST [1], participants viewed three stimuli that included two Japanese Hiragana characters each. The two characters for each stimulus were presented side-by-side in the display and formed the AB, CD, and EF pairs. Each character appeared equally often on the left and right side of the display across trials in the learning phase. These characters have been used in previous research with the PST task [1] and are expected to have low conceptual meaning to non-Japanese readers. Participants were instructed to select the character that was associated with correct feedback in each pair by pressing one of two keys on a keyboard (i.e., v-left, m-right). Feedback was presented after each choice. For the AB pair, selecting A led to positive feedback (i.e., “Correct!”) 80% of the time and selecting B led to negative feedback (i.e., “Incorrect”) 80% of the time. For the CD pair, selecting C led to positive feedback 70% of the time, and for the EF pair selecting E led to positive feedback 60% of the time. Individuals performed three learning blocks of 60 trials that included 20 trials for each of the pairs. In the transfer phase, participants viewed the eight novel pairs of the characters (i.e., A paired with CDEF, B paired with CDEF) four times each and were instructed to select the “better” of the two characters (i.e., the one more often associated with correct feedback in the learning phase), and feedback about individual’s choices was not presented.

The characters used in the PST were counterbalanced across the three feedback probabilities. In the learning and transfer phases, the characters were presented until either a response was made or until 4000 ms had elapsed if no response was given. For the learning phase, the feedback-to-stimulus interval was 1500 ms and the feedback remained on the screen for 1500 ms. The response-to-stimulus interval was 500 ms in the transfer phase. In the learning phase, the dependent variable was the proportion of trials where the preferred stimulus (ACE) was chosen for each of the three blocks. In the transfer phase, the dependent variables were the percentage of trials where A was selected or B was avoided (i.e., not selected). The probability of selecting A in the transfer phase provides a measure of learning from positive feedback, and the probability of avoiding B in the transfer phase provides a measure of learning from negative feedback.

### 2.3. Procedure

After participants arrived at the laboratory for the study, the application of the Electro-Cap (Electro-Cap International, Eaton, OH, USA) used to record the EEG data was described to the subjects, then an overview of the study was provided, and individuals signed the consent form. The protocol was approved by the Institutional Review Board of Iowa State University (Protocol 14-101: Learning and Decision Making, Approved 26 February 2014). Subjects completed a 10 item version of the Edinburgh Handedness Inventory [23], the Digit Symbol Substitution Test [24], the Mill Hill Vocabulary Test [25], and a short demographics questionnaire. Following the demographic and individual differences measures, the Electro-Cap was fitted to the subjects, who then completed the PST and a computerized gambling task while the EEG was recorded. The data for the gambling task are not reported in the manuscript.

### 2.4. EEG Recording and Analysis

The electroencephalogram (EEG) (bandpass 0.02–150 Hz, digitized at 500 Hz, gain 1000, 16 bit A/D conversion) was recorded from 68 tin electrodes based on a modified 10-10 system using an Electro-Cap (Electro-Cap International, Eaton, OH, USA). Vertical and horizontal eye movements were recorded from electrodes placed beside and below the right and left eyes. For recording the electrodes were referenced to electrode Cz; for data analysis the EEG was re-referenced to the average reference. A 0.1 to 20 Hz bandpass filter was applied to the data. Ocular artifacts related to blinks and saccades were compensated for using the Ocular Artifact Correction tool in EMSE 5.4 (Electromagnetic Source Estimation; Source-Signal Imaging, San Diego, CA, USA). The ERPs for the learning phase were averaged for correct and incorrect feedback trials for each of the three pairs from -200 ms before stimulus onset to 1500 ms after stimulus with a 200 ms prestimulus baseline. Trials contaminated by other artifacts (peak-to-peak deflections over 100 µV) were rejected before averaging.

The neural generators of the ERPs associated with feedback processing were examined using source analysis implemented in the Brainstorm software [26] running in Matlab 2013b. For the source analyses, a three shell spherical model of the cortex not including the brainstem or cerebellum was used along with the MNI152 template, minimum norm imaging, and sLORETA. In the “Surface” interface of Brainstorm, the “Min size” of the volume was set to 15 and the “Amplitude” to 25%. This served to reduce the contribution of spatially limited regions of activity to the activation maps. The MNI coordinates extracted from Brainstorm were converted to Talairach coordinates using the Sprout application (http://sprout022.sprout.yale.edu/mni2tal/mni2tal.html). The Talairach client (talairach.org) was used to determine names for regions of activation. The source analyses were conducted on the mean voltage of the incorrect minus correct ERP difference wave. The inputs to the source analysis represented the average of the 50 ms time windows used for the analysis of variance (ANOVAs) for the younger and older adults, respectively.

### 2.5. Data Availability

The behavioral and ERP data are available on the Open Science Framework at (https://osf.io/ane7s/).

## 3. Results

### 3.1. Behavioral Data

The learning phase data are presented in Table 2. These data represent the probability of selecting the A, C, or E stimuli, and were analyzed with a 2 (age: younger or older) × 3 (pair: AB, CD, EF) × 3 (block: 1–3) ANOVA. The main effect of age was not significant, *F* < 1.00, and neither were any of the interactions involving age (age × block *F* < 1.00; age × pair *F*(1,71) = 2.76, *p* = 0.067, *η_p_^2^* = 0.04; age × pair × block *F* < 1.00), these findings indicate that there were not age-related differences in the rate or level of learning in the learning phase. The main effect of block was significant, *F*(2,142) = 10.49, *p* < 0.001, *η_p_^2^* = 0.13, with selection of the A, C, and E stimuli increasing from blocks 1 to 2 (*t*(71) = 3.23, *p* = 0.006) and then stabilizing between blocks 2 and 3 (*t*(71) = 1.15, *p* = 0.76). The main effect of pair was significant, *F*(2,142) = 20.95, *p* < 0.001, *η_p_^2^* = 0.23, reflecting an increase from E to C (*t*(71) = 2.96, *p* = 0.012) and from C to A (*t*(71) = 3.48, *p* = 0.003). The pair x block interaction was not significant, *F*(4,284) = 1.92, *p* < 0.11, *η_p_^2^* = 0.03, indicating that the rate of learning across blocks was similar for the three pairs.

The data for the transfer phase are presented in Table 1. The effect of age was examined using a one-tailed independent *t-test* with the assumption that learning was greater for younger than older adults. The proportion of trials where A was chosen was not significantly different between younger and older adults, *t*(71) = 1.04, *p* = 0.15, d = 0.24. The proportion of trials where B was avoided was lower in older than in younger adults, and this difference was significant, *t*(71) = 1.99, *p* = 0.025, d = 0.47. This finding indicates that older adults were less likely to transfer learning from negative feedback to novel stimulus pairs than were younger adults.

### 3.2. ERP Data: Mean Voltage

The grand-averaged ERPs for younger and older adults representing the average of four electrodes over the medial frontal (FCz, Fz, FC1, FC2), temporal (left mastoid, right mastoid, FT9, FT10), parietal (CPz, Pz, P1, P2) and occipital (Oz, POz, O1, O2) regions are presented in Figure 1, and the mean voltage for the five components is presented in Table 3. These electrode locations were chosen based upon previous research from our laboratory examining the time course and topography of the ERP correlates of feedback processing [6,13,14]. The effects of age, feedback valence, and stimulus pair were quantified in a set of 2 (age: younger or older adults) × 2 (feedback: correct or incorrect) × 3 (pair: AB, CD, EF) × 4 (electrode) ANOVAs where significant age × feedback interactions were followed by test of the simple main effect of feedback for the younger and older adults. To compensate for age-related slowing of information processing [27], different epochs were used to measure the amplitude of the five components in younger and older adults: FN/temporal positivity (younger 250–300 ms, older 300–350 ms), frontal P3/occipital negativity (younger 350–400 ms, older 450–550 ms), parietal P3 (younger 325–375 ms, older 400–450 ms). Measurements for the components represent the average voltage across the 50 ms epoch for each individual.

#### 3.2.1. FN and Temporal Positivity

For the FN, the main effect of feedback was significant, *F*(1,71) = 47.78, *p* < 0.001, *η_p_^2^* = 0.40, reflecting greater negativity for incorrect than correct feedback (Table 3). The age × feedback interaction was significant, *F*(1,71) = 16.83, *p* < 0.001, *η_p_^2^* = 0.19, revealing an age-related decrease in the amplitude of the FN (younger adults, M = -1.78 μV, older adults M = −0.42 μV). The simple main effect test revealed that the FN differed for incorrect and correct feedback for both the younger adults, *F*(1,36) = 45.34, *p* < 0.001, and older adults, *F*(1,35) = 6.15, *p* < 0.018. This analysis also revealed a significant feedback x electrode interaction, *F*(3,213) = 4.30, *p* < 0.008, *η_p_^2^* = 0.06, with the effect of feedback being stronger at electrodes FC2 (*η_p_^2^* = 0.44), Fz (*η_p_^2^* = 0.34), and FC1 (*η_p_^2^* = 0.28), that at electrode FCz (*η_p_^2^* = 0.09); and a significant pair x electrode interaction, *F*(6,426) = 2.65, *p* < 0.019, *η_p_^2^* = 0.04, that was small and did not reveal a clear pattern of differences across the pairs.

For the temporal positivity, the main effect of feedback was significant, *F*(1,71) = 63.46, *p* < 0.001, *η_p_^2^* = 0.47, reflecting greater positivity for incorrect than correct feedback (Table 3); and the main effect of age was significant, *F*(1,71) = 6.55, *p* < 0.013, *η_p_^2^* = 0.08, reflecting greater negativity for older adults (M = -3.45 μV) than for younger adults (M = -1.71 μV). The age x feedback interaction was significant, *F*(1,71) = 22.21, *p* < 0.001, *η_p_^2^* = 0.24, with the amplitude of the temporal positivity being greater for younger adults (M = 2.34 μV) than for older (M = 0.57 μV) adults. The simple main effect test revealed that the temporal positivity differed for incorrect and correct feedback for both the younger adults, *F*(1,36) = 65.39, *p* < 0.001, and older adults, *F*(1,35) = 6.65, *p* < 0.014. None of the other main effects or interactions were significant.

#### 3.2.2. Frontal P3 and Occipital Negativity

For the frontal P3, the main effect of feedback was significant, *F*(1,71) = 44.22, *p* < 0.001, *η_p_^2^* = 0.38, reflecting greater positivity for incorrect than correct feedback (Table 3). The age × feedback interaction was significant, *F*(1,71) = 6.18, *p* < 0.001, *η_p_^2^* = 0.08, with the amplitude of the frontal P3 being greater for younger adults (M = 1.91 **μ**V) and for older adults (M = 1.00 **μ**V). The simple main effect test revealed that the frontal P3 differed for incorrect and correct feedback for both the younger, *F*(1,36) = 34.79, *p* < 0.001, and older, *F*(1,35) = 11.00, *p* < 0.002, adults. This analysis also revealed a significant feedback × electrode interaction, *F*(3,213) = 7.87, *p* < 0.001, *η_p_^2^* = 010, with the effect of feedback being stronger at electrodes FCz (*η_p_^2^* = 0.39), Fz (*η_p_^2^* = 0.41), and FC1 (*η_p_^2^* = 0.39) than at electrode FC2 (*η_p_^2^* = 0.25); a significant age x feedback x electrode interaction, *F*(3,213) = 4.74, *p* < 0.004, *η_p_^2^* = 0.06, where the age x feedback interaction was significant at electrodes FCz, Fz, and FC2, *F*’s (1,71) > 5.23, *p* < 0.025, *η_p_^2^* > 0.07, but not at electrode FC1, *F*(1,71) = 1.46, *p* = 0.23, *η_p_^2^* = 0.02; and a significant feedback × pair × electrode interaction, *F*(6,426) = 3.00, *p* < 0.01, *η_p_^2^* = 0.04, that was small and did not reveal a clear pattern of mean differences.

For the occipital negativity, the main effect of feedback was significant, *F*(1,71) = 82.39, *p* < 0.001, *η_p_^2^* = 0.54, reflecting greater negativity for incorrect than correct feedback (Table 3). In contrast to the frontal P3, the age x feedback interaction was not significant, *F*(1,71) = 1.82, *p* < 0.18, *η_p_^2^* = 0.03, indicating that the effect of feedback on the occipital negativity was similar in younger and older adults. This was supported by the simple main effect test as the effect of feedback was significant in the younger adults, *F*(1,36) = 42.57, *p* < 0.001, and older adults, *F*(1,35) = 42.24, *p* < 0.001. The feedback × pair interaction was significant, *F*(2,142) = 3.44, *p* < 0.037, *η_p_^2^* = 0.05, with the effect of pair being significant for correct feedback, *F*(2,142)= 3.45, *p* = 0.034, but not for incorrect feedback, *F* < 1.00.

#### 3.2.3. Parietal P3

For the parietal P3, the main effect of feedback was significant, *F*(1,71) = 79.69, *p* < 0.001, *η_p_^2^* = 0.53, the amplitude of the parietal P3 was greater for correct feedback than for incorrect feedback (Table 3). The age × feedback interaction was not significant, F < 1.00, *η_p_^2^* = 0.003, and the simple main effect test revealed that the effect of feedback on the parietal P3 was significant in younger adults, *F*(1,36) = 47.05, *p* < 0.001, and older adults, *F*(1,35) = 33.60, *p* < 0.001. Than main effect of pair was also significant, *F*(2,142) = 7.40, *p* < 0.001, *η_p_^2^* = 0.09, with the amplitude of the parietal P3 being greater for the CD (M = 2.19 μV), *t*(71) = 2.72, *p* = 0.024, and EF (M = 2.35 μV), *t*(71) = 3.71, *p* = 001, pairs than for the AB pair (M = 1.80 μV), and not differing between the CD and EF pairs, *t*(71) = 1.08, *p* = 0.85.

#### 3.2.4. Aging and Correct Feedback

From the data presented in Table 3, it appears that the effect of aging is primarily observed for incorrect feedback rather than for correct feedback for the FN/temporal positivity and frontal P3/occipital negativity. To quantify this observation, we conducted a set of ANOVAs including the data for correct feedback trials using both null hypothesis significance testing (NHST) and Bayesian methods. The Bayesian approach was included to examine the strength of the evidence for or against the null hypothesis. As can be seen in Table 4, the main effect of age was not significant in the ANOVAs for any of the four components and the Bayes 10 factor was below 1. These findings indicate that age did not affect the amplitude of the ERPs elicited by correct feedback trials in the PST.

### 3.3. ERP Data: Distributed Source Analysis

Distributed source analysis was used to explore the possible neural generators of the FN/temporal positivity and frontal P3/occipital negativity in the younger and older adults. Figure 2 portrays 2D topography maps for mean voltage of the incorrect feedback minus correct feedback difference wave for the FN and frontal P3 in younger and older adults. Figure 3 portrays the results of a set of distributed source analyses for younger and older adults.

The results of these analyses converged with the results of previous work using distributed source analysis to examine the neural generators of the FN and frontal P3 in a gambling task in younger [13] and older adults [6]. In the younger adults, the FN was associated with activity along the caudal section of the ACC (Figure 3a). This activity in the ACC was accompanied by bilateral activity within the midbrain. There was also bilateral activity in the lateral occipital lobe including the fusiform gyrus, and the middle temporal gyrus that appeared to be somewhat greater in the left hemisphere. This occipital and temporal activity may represent the temporal positivity. The frontal P3 was also associated with activity along the caudal section of the ACC that extended upward into the dorsomedial prefrontal cortex (Figure 3b). This time window also revealed activity within the posterior cingulate, posterior insula, and parahippocampal gyrus that may reflect the occipital negativity.

In the older adults, the FN was also associated with activity along the caudal section of the ACC (Figure 3a). During this time window there was also activity within the midbrain, and the lateral occipital cortex and middle temporal lobe. Additionally, one of the strongest regions of activity in the older adults was the medial frontal gyrus. This region also reveals activity in younger adults, but it was less pronounced relative to other regions. As was the case in the younger adults, the frontal P3 in the older adults was related to activity in the ACC, posterior cingulate, and posterior insula in the left hemisphere (Figure 3b). In contrast to the younger adults, the frontal P3 in the older adults was associated with only weak activity in the dorsomedial prefrontal cortex and was associated with activity extending over much of the anterior temporal lobe.

## 4. Discussion 

The FN and frontal P3 were associated with transient components of the ERPs over the temporal or occipital regions of the scalp, respectively. These findings are in agreement with previous work from our laboratory using a gambling task with ERPs [6,14], and converge with the functional neuroimaging literature demonstrating that feedback processing is associated with the recruitment of a distributed network that includes cortical and subcortical structures [17,18]. The FN and temporal positivity were related to activity in the ACC, lateral occipital cortex, and middle temporal gyrus; and the frontal P3 and occipital negativity were related to activity in the ACC, posterior cingulate and insula. These findings lead us to suggest that the ACC may receive input from a variety of cortical and subcortical structures between 200 and 500 ms after feedback is received indicating the outcome of a choice. Given similarities between the current findings and those of our previous research using a gambling task, interactions between the ACC and these posterior cortical structures may represents a rather general feedback processing network that transcends the specific demands of a given task [16,28]. One avenue for future research could be to compare the ERP correlates of feedback processing in two or more tasks within the same participants to explore the overlap and separability of the neural networks underpinning feedback processing related to gambling and reinforcement learning.

The amplitude of the FN and temporal positivity was attenuated in older adults. With regard to the FN, this finding converges with evidence from previous studies using both reinforcement learning and gambling tasks [6,19,21]. In the distributed source analysis, the FN and temporal positivity were associated with similar neural generators in the younger and older adults that were located in the ACC, occipital, and temporal cortices. The most notably difference in the source analysis of the FN/temporal positivity between the two groups represented the relative difference in the strength of activity within the medial frontal gyrus, being stronger relative to other regions in older adults, but not in younger adults. Together, the results of the mean voltage and distributed source analyses lead us to suggest that aging is associated with a reduction in the connectivity between the ACC and posterior cortical structures (i.e., occipital cortex, middle temporal gyrus) underpinning the generation of the FN and temporal positivity, and possibly the up-regulation of more local connections between the ACC and medial frontal gyrus. This suggestion converges with evidence from the functional neuroimaging literature demonstrating a reduction of posterior-to-anterior connectivity coupled with increases in frontal lobe activity in later adulthood across a variety of tasks [29].

The amplitude of the frontal P3 was attenuated in older adults, a finding that differs from previous research demonstrating age-related invariance of the frontal P3 in learning [22] and gambling [6] tasks. The current finding may reveal a primary deficit within the ACC related to the frontal P3. The reason for difference between the current finding of an age-related decrease in the amplitude of the frontal P3 and the age-related invariance of the frontal P3 in two previous studies is unclear. The effect of age on the FN was larger than the effect of age on the frontal P3 in the current dataset, and the sample size of the previous studies that reported a non-significant effect of age on the frontal P3 was smaller (e.g., West et al. [6] included only 20 younger and 20 older adults) than the current study. Given this, it is possible that prior studies failed to detect a small effect of aging on the frontal P3 given the limited sample size.

In contrast to the effect of aging on the frontal P3, the effect of feedback on the occipital negativity did not differ significantly in young and older adults. This finding may indicate that activity within the posterior cingulate and insula around the time of the frontal P3 is preserved in later adulthood. In contrast to the FN and temporal positivity, the effect of aging during the time interval of the frontal P3 may reflect a local reduction of activity within the ACC and dorsomedial prefrontal cortex, rather than a general dampening of activity across a network including the ACC, posterior cingulate, and insula. Variation in the effect of aging on the frontal P3 and occipital negativity may be consistent with some data from structural MRI demonstrating that age-related reductions in gray matter volume are observed earlier in anterior cingulate (BA24) than in some posterior cingulate regions (BA23, BA29) [30].

The amplitude of the ERPs elicited by positive feedback was insensitive to the effect of aging, and learning from positive feedback was similar in younger and older adults. The general absence of age-related differences in the processing of positive feedback is consistent with the principles of Socioemotional Selectivity Theory [31,32], wherein there is believed to be a shift towards a focus on positive outcomes or life experiences in later adulthood for healthy individuals. In our data, this shift may contribute to the preserved processing of, and learning from, positive feedback in the PST observed in the older adults.

The ERPs measured during the learning phase and the behavioral data measured during the transfer phase reveal an age-related reduction in the processing of negative feedback, and a decrease in the ability of older adults to generalize learning from negative feedback to guide decision-making for novel stimulus pairs. The effect of aging on the amplitude of the FN is consistent with evidence from a number of studies using various paradigms to examine the effect of aging on the neural correlates of feedback processing [6,19,20,21,22], and may therefore reveal a rather general age-related reduction in the processing of negative feedback. The dissociation between the learning phase data, wherein age-related differences in learning were not observed, and the transfer phase data, wherein older adults did not express learning from negative feedback may seem curious. In their computational modeling work, Frank et al. [1] have demonstrated that two types of processes contribute to performance in the learning phase of the PST. Processes related to “working memory” may serve to guide trial-to-trial choices during the learning phase, while associative processes that integrate information over trials may support reinforcement learning. Within the context of this distinction, it is possible that working memory processes were sufficient to support choices in the learning phase in older adults where there were only three stimuli to track over time. In line with this idea, the older adults did demonstrate greater frontal recruitment, particularly in the analysis of the FN [29]. In contrast, age-related declines in associative processes driven by negative feedback may be revealed in the transfer phase when the task required using prior learning to make inferences about novel pairings of the stimuli that could be related to recruitment of the parahippocampal gyus in the analysis of the frontal P3 and occipital negativity in the learning phase [33,34].

The amplitude of the parietal P3 was greater for correct feedback than for incorrect feedback, was sensitive to stimulus pair, and appeared to be insensitive to the effect of aging. The effect of feedback valence on the parietal P3 converges with the findings of other studies examining this component in the context of feedback processing [8,15,35]. The different effects of feedback on the parietal P3 and frontal P3 components, indicates that it is important to distinguish between these two components when studying feedback processing. The effect of stimulus pair on the parietal P3 is consistent with other evidence indicating that the P3b is sensitive to the expectancy of a stimulus or outcome [35,36], and in the PST this expectancy appears to be driven by prior experience or learning given differences in the amplitude of the component between the AB, and CD and EF pairs. The absence of an effect of aging on the parietal P3 may be surprising given the broader literature that typically reveals that the amplitude of the P3b is reduced in older adults, accompanied by differences in topography of the component between younger and older adults [37,38]. However, there is some evidence indicating that the P3b is relatively insensitive to aging when the component is separated from frontal ERPs related to novelty processing [39]. This may account for the lack of age-related differences in the parietal P3 in the current study as the learning phase only included three stimuli that were equally probable, and the older adults seemed to have learned the probabilistic associations related to the stimuli as well as the younger adults in the learning phase.

## 5. Conclusions

The current findings converge with prior research from our laboratory and the neuroimaging literature in demonstrating that feedback processing is related to a network including the ACC, and regions within the temporal and parietal cortex between 300 and 500 ms after feedback is presented [13,18]. Based upon the current data, it seems that there are at least two different effects of aging on the ERP correlates of feedback processing in this study: (1) A local reduction in feedback processing within the ACC, reflecting the reduction in the amplitude of the FN and frontal P3 in older adults, and (2) a weakening of connectivity between the ACC and posterior cortical structures (i.e., occipital, middle temporal gyrus, posterior cingulate, insula), reflecting in the effect of aging on the FN and temporal positivity.

## Figures and Tables

**Figure 1 brainsci-10-00040-f001:**
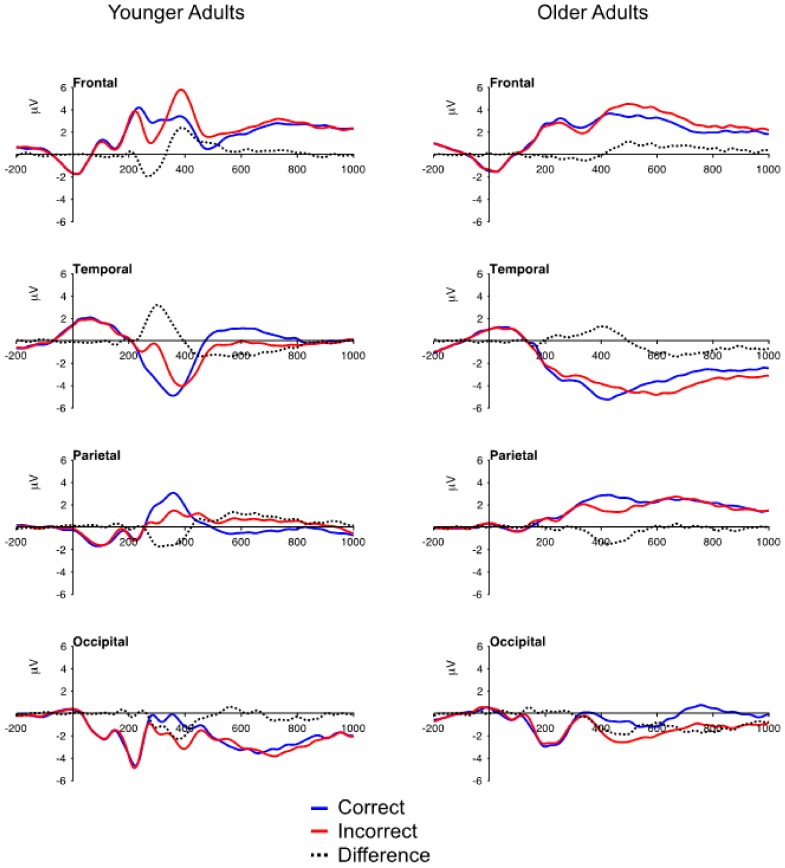
Grand-averaged ERP waveforms for correct and incorrect feedback trials and the incorrect minus correct difference waves for younger and older adults averaged across four medial frontal, temporal, parietal, and occipital electrodes. The tall bar marks onset of the feedback stimulus and positive is plotted up.

**Figure 2 brainsci-10-00040-f002:**
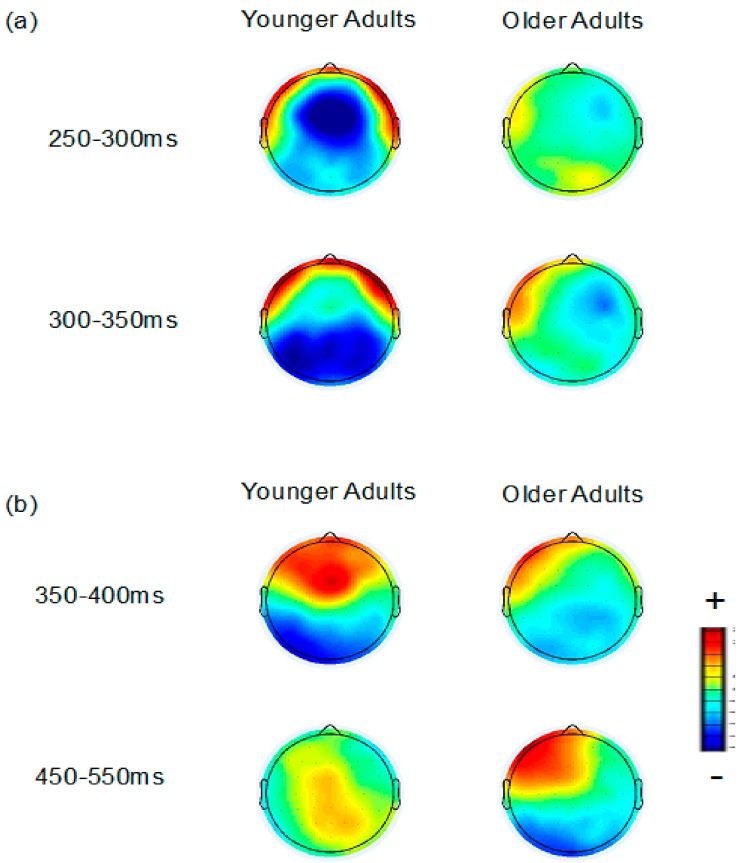
2D topography maps for incorrect minus correct feedback demonstrating the distribution of the (**a**) FN and (**b**) frontal P3 in younger and older adults. Blues represent greater negativity for incorrect than correct feedback and reds represent greater positivity for incorrect than correct feedback.

**Figure 3 brainsci-10-00040-f003:**
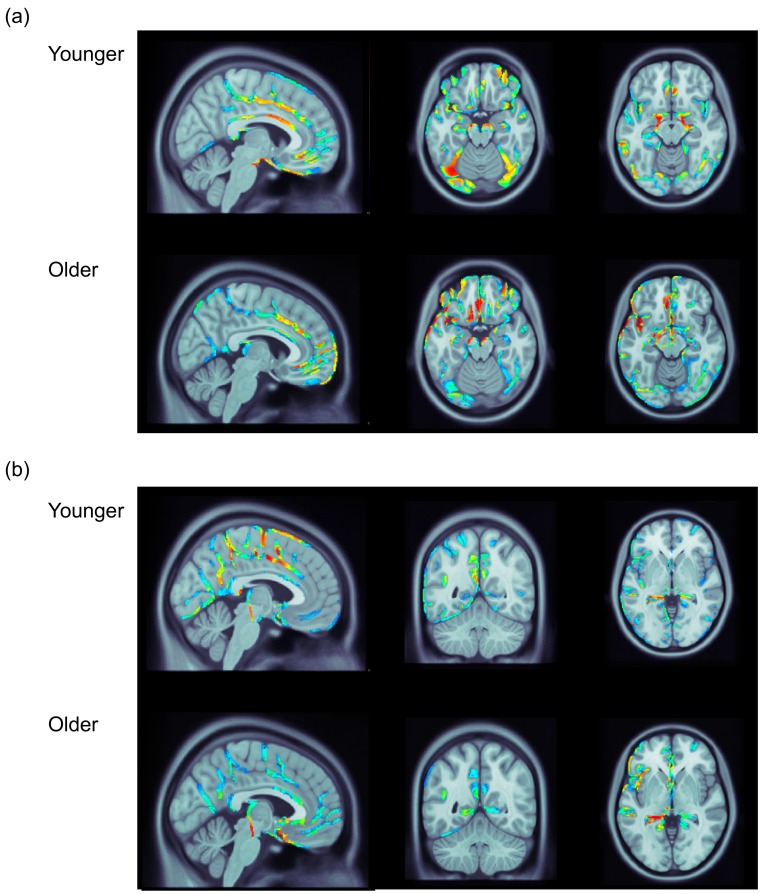
Results of the distributed source analysis for younger and older adults at the peak of the components for the relevant group. (**a**) Results for the FN, with the left panel demonstrating activity along the ACC, the middle panel demonstrating activity in the occipital region in both groups and the medial frontal region in older adults, and the right panel demonstrating activity in the middle temporal region. (**b**) Results for the P3a, with the left panel demonstrating activity along the ACC, and dorsomedial frontal cortex in younger adults, the middle panel demonstrating activity in the posterior cingulate and posterior insula, and the right panel demonstrating activity in the parahippocampal region. Moving from darker blue to red represents increasing activity.

**Table 1 brainsci-10-00040-t001:** Means (M) and standard deviations (SD) for demographic, psychometric, and test phase data for younger and older adults.

		Younger	Older
Age	M	19.56	71.42
	SD	1.56	5.79
Education	M	13.94	17.36
	SD	0.95	2.76
Vocabulary	M	14.62	23.14
	SD	3.60	4.80
Digit symbol	M	43.08	32.06
	SD	6.01	7.30
Choose A	M	0.66	0.60
	SD	0.22	0.23
Avoid B	M	0.63	0.54
	SD	0.19	0.21

**Table 2 brainsci-10-00040-t002:** Learning phase data (mean (M) and standard deviation (SD)) for younger and older adults.

		A		B		C	
		Younger	Older	Younger	Older	Younger	Older
Block 1	M	0.70	0.66	0.65	0.57	0.51	0.57
	SD	0.23	0.21	0.18	0.18	0.19	0.20
Block 2	M	0.72	0.72	0.68	0.65	0.55	0.60
	SD	0.21	0.25	0.19	0.19	0.18	0.21
Block 3	M	0.78	0.78	0.68	0.61	0.55	0.60
	SD	0.17	0.19	0.21	0.22	0.18	0.17

**Table 3 brainsci-10-00040-t003:** Mean voltages in microvolts and standard deviation for younger and older adults for correct and incorrect feedback trials related to the FN, frontal P3, temporal positivity (temp. pos.), occipital negativity (occ. neg.) and parietal P3.

		FN	Frontal P3	Temp. pos.	Occ. neg.	Parietal P3
Younger Adults						
Correct feedback	M	3.18	3.33	−2.88	−0.57	2.87
	SD	2.52	2.44	2.57	2.56	2.31
Incorrect feedback	M	1.40	5.24	−0.54	−2.43	1.29
	SD	2.43	3.21	1.71	2.97	2.75
Older Adults						
Correct feedback	M	2.44	3.43	−3.73	−0.87	2.85
	SD	3.38	3.43	3.68	3.80	2.76
Incorrect feedback	M	2.02	4.43	−3.16	−2.39	1.43
	SD	3.55	3.68	2.99	4.24	3.22

**Table 4 brainsci-10-00040-t004:** F-ratio, *p*-values, and the Bayes 10 (BF_10_) factor for the main effect of age group in the comparison of “correct” feedback trials.

	F	*p*	BF_10_
FN	1.09	0.30	0.60
Temporal positivity	1.61	0.21	0.57
Frontal	0.05	0.83	0.38
Occipital negativity	0.23	0.64	0.37

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
