# Peer review of "The Effect of Aging on the ERP Correlates of Feedback Processing in the Probabilistic Selection Task"

_brainsci, 2020, doi:10.3390/brainsci10010040_

Round 1

Reviewer 1 Report

Review of the article “The Effect of Aging on the ERP Correlates of Feedback Processing in the Probabilistic Selection Task” submitted by Robert West and AnnMarie Huet for publication in Brain Sciences.

Summary

The authors present an article aimed at examining the effect of aging on ERP correlates of feedback processing in a probabilistic learning task. The first goal was to replicate and extend findings from a gambling task showing that the feedback negativity (FN), but not the frontal P3, is attenuated in older adults to a reinforcement learning task. The second goal was to assess the impact of aging on additional ERP components associated with more posterior brain regions that covary in time with the FN and frontal P3. The results showed that the FN and frontal P3 were attenuated in the older adults. A temporal positivity was also attenuated in aging, but an occipital negativity was not.

General Overview

This study is focused on a timely and important question about age differences in the neurophysiological underpinnings of reinforcement learning. The experiment appears well designed and the analyses are thorough. The results are relatively convincing, but there are a number of points that should be clarified. I have provided a number of specific comments below.

Specific Comments

Introduction

The introduction is generally well written and includes a literature review of appropriate breadth to introduce the research question.

Pages 2-3 – The motivation for the first hypothesis about the effect of aging on the FN and frontal P3 was clear. However, the hypothesis that aging is associated with a decrease in the amplitude of components temporally contiguous with the FN, but not with the frontal P3 was less clear. Perhaps some additional rationale for these specific hypotheses would be helpful.
Page 1-2 – the ACC is first mentioned on line 37, but the acronym is not defined until line 51. Please move this definition up to line 37.

Methods

Page 3 – it was not entirely clear how the chi-squared test was conducted, and what the significant result entails. It appears that it tested the substantial difference between the age groups in the sex distribution of the samples. Did the authors us sex as a covariate in any of the analyses? If not, would the results change with the covariate? Also, there is a superscript 1 at the end of the first sentence, but there was no corresponding footnote (as far as I could tell).
Page 3 – it was not clear where the participants were recruited from, or where the study was conducted. This comment was partly driven by curiosity about the age difference in levels of education, but also driven by the fact that the IRB was from Iowa State, but the authors are not currently there.
Page 3, lines118-119 – the procedure for the “test” phase were not entirely clear. What did the authors mean by “the six novel pairings of the characters”? I assume that the six characters were recombined into novel pairings (relative to the pairings in the learning phase), but this was not explicitly stated. Please clarify.
Page 3, line 122 – A brief statement to clarify that all stimuli were Japanese Hiragana characters (rather than English ABCDEF characters) would be helpful for the reader. Plus, a statement of the rationale for using these characters would be beneficial. Related, were all participants native English speaking, non-Japanese speaking individuals?
Page 4, line 139 – the authors mentioned that participants performed a computerized gambling task after completing the PST, but there are no data presented from this gambling task, and as far as I noticed it was not mentioned again.

Results

Page 5 – why were the mastoid electrodes included in the temporal region grand averages?
Line 204 – There appears to be a typo here with the word “adults” at the end of the sentence.

Discussion

Page 10, lines 323-326 – The authors note that there was an attenuation of the frontal P3 in the current study, contrary to previous studies. Is there a reason why ACC activity might be reduced in older adults in a probabilistic learning task, but not in a spatial learning, or gambling task? The differences between these findings perhaps warrants further discussion.
General comment – I think the paper would benefit from some additional discussion that tries to relate the ERP findings to the behavioral findings. In the learning phase of the experiment younger and older adults were equally able to accrue information about the feedback probabilities in order to make judgments about which member of each stimulus pair was associated with correct feedback. Despite the fact that younger and older adults were equally able to use feedback probabilities to drive behavioral responses, older adults had an altered neurophysiological response to the negative and positive feedback. It is not until the test phase that this altered response has an impact on behavior, and even then, the diminished response to positive feedback did not seem to affect behavior. Is the altered response to the feedback related to memory processes (e.g., encoding) of the feedback, perhaps into long-term memory? Are older adults forgetting the feedback they received during the learning phase, thereby affecting their performance during test? Could this be related to the parahippocampal activation? Did the groups differ in this parahippocampal source? Are there other potential interpretations given the specific networks revealed in the source analysis?

Author Response

1. Pages 2-3: The reviewer suggested that we could strengthen the motivation for the hypotheses related to the temporal negativity and the occipital positivity.

In lines 98-122 we have tried to provide a stronger motivation/justification for the hypotheses related to the temporal negativity and occipital positivity. Our ideas are primarily taken from the observed effects of age on the FN and frontal P3, as there is not prior research examining the effects of aging on the non-medial frontal ERP components beyond West et al. 2014.

2. Pages 1-2: The reviewer suggested that we should move the name and acronym for the anterior cingulate cortex.

In line 37 use spell out “anterior cingulate cortex” and included the acronym.

3. Page 3: The reviewer suggested that the structure of the chi-square test was not clear and ask whether sex was used as a covariate in the analyses?

In lines 132-133 we have tried to clarify the structure of the chi-square analysis indicated that it included factors sex and age. We have not included the variable sex as a covariate in the analyses. Given that sex effects on cognitive variables tend to be relatively small and to be quantitative rather than qualitative in nature, we doubt that including sex as a covariate in the analyses would change then results in a substantial manner. We may be underestimating age-related differences to the degree that the effect of aging is sometimes reported to be greater for men than women, but again this effect tends to be small so a larger sample would be needed to consider its impact.

4. Page 3: The review indicated that it was not clear how the participants were recruited.

In lines 128-130 we have added a statement indicating that younger adults were undergraduates at Iowa State University and that older were community dwelling individuals recruited from a lifelong learning program affiliated with the university. The data were collected while I was on faculty at Iowa State and the second author was an undergraduate student. We are happy to add this information to the author affiliations if this is the practice of the journal.

5. Page 3, lines 118-119: The reviewer ask for some additional information to clarify the structure of the PST.

In lines 143-148 and 199-201 we have sought to refine the description of the PST so that the structure of the task and stimuli are more clearly presented.

6. Page 3 line 122: The reviewer ask for a statement related to the rationale for using the Hiragana characters and queried whether the participants were native English speakers.

In lines 147-148 we include a statement that the characters used in the task were taken from work by Frank et al. [1] with the PST that we modeled the task from. All of the participants were fluent in English. We unfortunately did not ask about experience with Japanese, although given the student population and the population of Ames, IA we doubt that any of the subjects were native Japanese speakers, and to our recollection this was not mentioned during the testing sessions by any participants.

7. Page 5: The reviewer asked why the mastoid electrodes were used in the analysis for the temporal positivity. 

On lines 339-341 we have added a sentence indicating that the electrodes used in the analyses of the five ERP components were based upon our previous research. The mastoid electrodes are the closest electrodes for Ft9-Ft10 in the montage used for recording the data. We suspect that this question may arise from the convention in some areas of ERP research to use the mastoid as the reference electrode. 

8. Page 10 lines 323-326: The reviewer ask where there is any reason that the current findings related to the frontal P3 might differ from the published research. 

On lines 521-527 we have sought to address this comment. One possibility is related to the power of the studies. For instance, West et al. [6] included 20 younger and 20 older adults. The effect of age was smaller for the frontal P3 than for the FN in the current data, so it seems possible that prior studies may have failed to detect a small effect of age on the frontal P3.

9. General comment: The reviewer suggested that the paper would benefit from an attempt to better integrate the findings of the behavioral and ERP data.    

In lines 538-617 we have tried to address this comment. In the paragraph running from lines 545-617 we describe the ideas of Frank et al. where different processes are thought to contribute to trial-to-trial choices and reinforcement learning in the PST. Differences in the sensitivity of these processes to aging may account for the pattern of age-related differences and stability observed in our study.

Reviewer 2 Report

In this study the authors investigate the effect of aging on feedback processing using a multichannel-EEG acquisition in ad-hoc experiment involving young and old adults. This is a well conducted study and clear manuscript. The discussion and interpretation are justified by the results; however there are a few concerns which need clarification.

First of all:  how did the authors classify the age group? I mean the range of each group (btw there is only the mean age in Table 1).  Please justify in the methods using reference or reasonable motives.

Abstract: what do you mean for “longue range interactions”. Could you be more precise?

Methods- EEg acquisition. I don’t understand the sentence: “The volume size was set to 15 and the amplitude to 25% to reduce the  presence of spatially limited regions of activity in the activation maps.” Could you pleasebe more clear about that?

Methods- did the authors excluded the cerebellum from the model they used to compute the source localization? Please explain.

Figure 1, the unit on the y axis is missing, please provide it.

The caption of Table 3 is not clear. Does “mean voltages” indicate the maximum amplitude from 0V or is the amplitude measured pick to pick? Units are also missing.

Section 3.3 source analysis. It is not clear to me what time range they used for each source analysis. Please clarify.

Figure 2. please report the time range within the source analysis was evaluated. Also the color-coded brain activity is not clear. What does it represent? Please add a legend and explain it in the caption.

Discussion: can we speculate that aging modify/reduce the connettivity between the involved areas? Please discuss it in the light of a similar results; for example  the effect of aging in taste perception see the paper: (Age-related changes of gustatory function depend on alteration of neuronal circuits, J Neuro Res. 2017;1–10.)

Conclusions: for “weakening of interactions” do you mean “weakening of connectivity”? please clariy and eventually re-phrase it.

Author Response

The reviewer suggested that we should report the age range of the two groups.

In line 131 we report the age range for the older and younger adults. The selection of the age range follows prior research from my laboratory and the cognitive aging/neurocognitive aging literature more generally. We had a lower limit in the older adults of 60 years (63 realized) and don’t tend to see a lot of healthy community dwelling older adults able/willing to come to the lab to participate in research beyond the mid 80’s.

The reviewer suggested that we should be more precise in our meaning of interactions.

In line 22 we have switched to the suggested term “connectivity”, we had avoided this terms in the initial submission since we did not conduct formal test of connectivity given the short measurement windows that may not be appropriate for EEG connectivity analyses.

The reviewer suggested that the structure of the source analysis needed to be more clearly described related to this volume size and amplitude threshold; and ask whether the cerebellum was included in the source model.

In lines 233-245 we have sought to refine the description of the structure and method for the source analysis. The cerebellum was not included in the model.

The units have been added to the y-axis for Figure 1.

We have refined the title for Table 3 and state that the units are microvolts. Related to this we have corrected places in the text where “mu” had been converted to “m”.

In lines 243-245 we state that the input for the source analyses represented the average of the 50 ms epochs used in the analyses of mean voltage.

We have updated the legend for Figure 3 to indicate the meaning of the colors presented in the figures.

We have tried to enhance the discussion section related to the source analysis (Reviewer 1 point 9). We have not introduced the paper on gustatory processing noted by the reviewer as the topic explored in this paper seems to be well outside of the questions explored in our paper.